# Tobacco Smoke Exposure According to Location of Home Smoking in Israel: Findings from the Project Zero Exposure Study

**DOI:** 10.3390/ijerph20043523

**Published:** 2023-02-16

**Authors:** Laura J. Rosen, David M. Zucker, Shannon Gravely, Michal Bitan, Ana M. Rule, Vicki Myers

**Affiliations:** 1Department of Health Promotion, School of Public Health, Sackler Faculty of Medicine, Tel Aviv University, Tel Aviv 6997801, Israel; 2Department of Statistics and Data Science, Hebrew University, Jerusalem 9190501, Israel; 3Department of Psychology, University of Waterloo, Waterloo, ON N2L 3G1, Canada; 4Department of Computer Science, College of Management Academic Studies, Rishon LeTsiyon 7579806, Israel; 5Department of Environmental Health and Engineering, Johns Hopkins Bloomberg School of Public Health, Baltimore, MD 21205, USA; 6Gertner Institute of Epidemiology and Health Policy Research, Sheba Medical Center, Ramat Gan 5262100, Israel

**Keywords:** children, tobacco smoke exposure, home-smoking, hair nicotine, biomarker, home smoking rules

## Abstract

Young children are particularly vulnerable to harms from tobacco smoke exposure (TSE). This study aimed to compare TSE: (1) between children who live in smoking families and those who do not; and (2) among children who live in smoking households with varying smoking locations. The data came from two studies that were conducted concurrently in Israel (2016–2018). Study 1: a randomized controlled trial of smoking families (*n* = 159); Study 2: a cohort study of TSE among children in non-smoking families (*n* = 20). Hair samples were collected from one child in each household. Baseline hair nicotine data were analyzed for 141 children in Study 1 and 17 children in Study 2. Using a logistic regression analysis (exposed vs. not exposed as per laboratory determination) and a linear regression (log hair nicotine), we compared TSE between: (1) children in Study 1 vs. Study 2; (2) children in families with different smoking locations in Study 1: balcony; garden, yard, or other place outside of the home; or inside the home (designated smoking areas within the home (DSAs) or anywhere). A higher proportion of children living in smoking households were measurably exposed to tobacco smoke (68.8%) compared to children living in non-smoking households (35.3%, *p* = 0.006). Among children from smoking families, 75.0% of those whose parents smoked in the house were exposed, while 61.8% of children whose parents restricted smoking to the porch (*n* = 55) were exposed, and 71.4% of those whose parents smoked outside the home (including gardens and yards) (*n* = 42) were exposed. In univariable and multivariable models, smoking location was not significantly associated with exposure. The majority of children in smoking families were measurably exposed to TSE, even if smoking was restricted to designated areas in the home, balconies, orgarden/yard/other outdoor areas. Reducing population smoking rates, particularly among parents, restricting smoking to at least 10 meters from homes and children, and denormalizing smoking around others are recommended to reduce population-level child TSE and tobacco-attributable disease and death.

## 1. Introduction

Tobacco use continues to be the leading preventable cause of premature illness and death in the world. Tobacco kills up to half of those who use it, resulting in over 7 million global deaths each year. An additional 1.2 million people die from diseases caused by exposure to secondhand smoke (SHS) [1]. Infants and young children exposed to tobacco smoke are at an increased risk of a range of diseases, including low birth weight, sudden infant death syndrome, ear infections, asthma, pneumonia, and inflammatory effects leading to metabolic syndrome, insulin resistance, impaired cardiac autonomic function, and premature atherosclerotic heart disease [2,3,4]. At least 500 million children worldwide are exposed to secondhand smoke at home [5]. 

Scientific evidence showing the causal relationship between tobacco smoke exposure (TSE), which includes both second- and third-hand exposure, and death and disease, has led to a proliferation of smoke-free laws in public places throughout the world, particularly in bars, restaurants, hospitality venues, and workplaces [6,7]. Article 8 of the World Health Organization Framework Convention on Tobacco Control (FCTC) obligates parties to adopt and implement legislative and other measures providing for protection from exposure to tobacco smoke in indoor public places, workplaces, public transport, and, as appropriate, other public places [8]. Smoking bans can result in an overall 40% reduction in SHS exposure, up to an 80–90% reduction in high-exposure areas, and reduce illness, death, and healthcare costs [9,10]. However, at this time, the FCTC guidelines do not cover smoking laws inside and around personal dwellings, particularly for multiunit housing where involuntary exposure to secondhand smoke most frequently occurs [2,11]. Some parents who are aware of the potential dangers of exposing their children to smoke voluntarily take precautions to protect them. A study conducted in Hong Kong showed that maternal protective strategies included opening windows, asking the father not to smoke near the child, moving the child away from the smoke, and removing ashtrays [12]. In our qualitative study of smoking parents in Israel, we found that parents in smoking families employed various strategies with different levels of restrictiveness, ranging from never permitting smoking inside or around the home, to smoking on the outside balcony with the door open or closed, to smoking only on an indoor balcony or designated areas in the home, to smoking “in” or at a window, or to smoking only when children are not present in the home. The parents themselves were often confused about the benefits of these strategies [13,14].

Restricting smoking to outside of the home has long been thought to help reduce and even eliminate a child’s exposure to tobacco smoke. However, studies have shown that even when parents restrict smoking to areas outside the home, exposure may occur, increasing the risk of disease for children. A study in California by Matt et al. found that young child TSE, based upon cotinine levels, was 5–7 times higher in households of smokers who smoked outdoors than in households of nonsmokers [15]. Yamakawa et al. discovered that maternal indoor and outdoor smoking were both associated with an increased risk of child hospitalization for respiratory tract infection [16]. A study in Sweden by Johansson et al., including young children (ages 2.5–3 years), found that parents who smoked outdoors (with the door closed) still significantly exposed their children to twice the odds of TSE relative to children living in non-smoking households [17]. Jurado found that both smoking in the living room and smoking on the porch were associated with increased child urinary cotinine levels [18]. A study of nursery school children in Germany found that home smoking bans had a protective effect [19]; the study did not differentiate between smoking on porches, smoking on terraces, and smoking in gardens. Al Delaimy et al., in a study conducted in New Zealand, found no differences in children’s hair nicotine levels as a function of indoor versus outdoor smoking, without reference to balcony smoking [20]. A study conducted in Turkey found no statistically significant differences in urinary cotinine levels among asthmatic children whose parents smoked indoors versus outdoors [21]. 

Smoking rates are high in Israel (1 in 5 adults smoke), and over 60% of children are exposed to TSE [22]. However, little is known about the association between the location of smoking around the home and child exposure. 

We aimed to (1) assess TSE exposure in study samples of children living in non-smoking and smoking families in Israel and (2) assess whether TSE differed by smoking location (inside, on the balcony/porch, or in the yard, garden, or other place outside of the home). We hypothesized that an unrestricted smoking policy, where smoking is allowed in any room or part of the home, would be associated with the highest levels of TSE; however, based on previous literature, we also anticipated detectable levels of TSE among children whose parents restricted smoking to balconies or yards. 

## 2. Materials and Methods

### 2.1. Study Design, Samples and Procedures

The data came from two studies that were conducted in Israel at the same time between 2016 and 2018. The first study (Study 1) was a randomized controlled trial (RCT), “Project Zero Exposure,” which included 159 families with at least one parent who smoked cigarettes living in the household. The RCT assessed an intervention program to help parents protect their children from tobacco smoke. Details about Project Zero Exposure can be found elsewhere [23,24,25]. The second study (Study 2) was a cohort study of 20 non-smoking families, where both the mother and the father did not smoke [26]. For both studies, recruitment was conducted via daycare centers [27], via social media parents’ groups, billboards, and the “snowball” method. Baseline data collection for both studies was identical. Families who completed either study received a gift certificate worth $70 (USD) as compensation for their time. [23] Recruitment was conducted in various regions of Israel.

### 2.2. Eligibility Criteria

Inclusion criteria included: (1) at least one smoking parent living in the household who smoked at least 10 cigarettes per week (Study 1) or no parents who smoked in the household (Study 2); (2) having a child aged up to 8 years; (3) the parent(s) were willing to provide a hair sample for research purposes; and (4) children had a sufficient hair sample for the laboratory analysis. 

### 2.3. Ethics and Registration

National Institute of Health (NIH) Clinical Trials Registry: NCT02867241. Ethical approval was obtained from the Helsinki Committee of Asaf Harofe Hospital (0143-16 ASF), the Israel Ministry of Health (920090057), and the Tel Aviv University Ethics Committee. All parents provided written consent for their children to participate.

### 2.4. Measures

#### 2.4.1. Primary Outcome

The primary outcome for this analysis was child exposure to tobacco smoke, measured by hair nicotine. Child hair nicotine was chosen because it reflects long-term exposure (approximately 1 cm of hair growth/month) [28], it is noninvasive [29], it is easily stored and transported, and it is stable for long periods of time. Hair nicotine measures exposure to both secondhand smoke (smoke emitted from a burning cigarette or from a smoker’s mouth) and thirdhand smoke (the aged residual nicotine and other chemicals found in the atmosphere and on surfaces after the smoker has left or extinguished the cigarette). Hair samples were collected from the back of the head, as close to the scalp as possible, from one child in each family. Analysis was performed by gas chromatography with mass spectrometer detection (GC-MS) at Johns Hopkins University [30]. Prior to analysis, hair samples were washed using laboratory procedures to reduce the impact of nicotine found on the hair surface. Samples with low hair mass (<10 mg) were excluded. The data were analyzed in 10 laboratory batches. Each child was categorized as “exposed” or “not exposed” based on the Limit of Detection (LOD), which was calculated for each batch separately based on the average LOD among samples in that batch. Each individual sample was deemed above or below the batch-specific LOD. We report hair nicotine values as nicotine ng/mg.

We analyzed TSE in two ways. First, we assessed TSE as a binary variable indicating “exposed” (yes, above the limit of detection) versus “unexposed” (no, below the limit of detection). This approach is useful because there is no safe level of exposure to tobacco smoke, so any exposure is important, and also because it provides information on the proportion exposed, which is easily understandable. Second, we assessed TSE as a continuous variable based on the levels of hair nicotine and used the natural log transformation because of the known non-normal distribution of hair nicotine. Hair nicotine was analyzed in ten separate batches over the course of the research.

#### 2.4.2. Independent Measures

Study 1 vs. Study 2: The independent variable for the first analyses was whether the children were from smoking or non-smoking families. 

Study 1: The independent variable for the second assessment was the location of smoking. We asked parents to show us where they usually smoked, with the question “Where does smoking usually take place?” and recorded and categorized the responses into four categories: (1) garden/yard/other place outside of the home; (2) balcony (without differentiating between degree of enclosure of the porch or whether the balcony had a roof); (3) designated smoking areas inside the home (DSA) (e.g., enclosed service/laundry balcony/clearly indoor balcony, designated room, or smoking at the window); or (4) anywhere inside the house. Because only 3 families reported smoking in the entire house, we combined categories 3 and 4 into a single category, “in the home,” for the purpose of statistical analysis. 

We asked about the daily number of cigarettes smoked by the parents and created a variable that indicated the combined number of cigarettes smoked daily. We asked about which parent smokes (mother only, father only, both), and categorized the variable into two categories: (1) Both parents smoke; or (2) only one parent smokes.

Sociodemographic questions: sociodemographic data were used in adjusted models. Data collected were: child age in months; parental nationality (both parents Israel-born/other); maternal education (categorized into non-academic and academic; at least some university-level education); paternal education (categorized into non-academic and academic); monthly household income (categorized into above average, average, and below average); neighborhood SES (scale of 1–20, with 20 the highest) [31].

### 2.5. Statistical Analyses

Descriptive statistics were used to describe the study samples. 

#### 2.5.1. Analyses, Smoking Families vs. Nonsmoking Families (Study 1 vs. Study 2)

We compared child TSE (above the LOD: yes vs. no) between non-smoking and smoking families using a chi-squared test. We used a linear model to compare child exposure between non-smoking and smoking families as measured by log-hair nicotine (LHN) while controlling for laboratory batch. Because of the small number of participants in the non-smoking family cohort, we did not perform multivariable analyses. 

#### 2.5.2. TSE by Location of Home Smoking and Other Explanatory Variables (Study 1)

We first assessed TSE by location of home smoking using the LOD (dichotomous outcome: exposed vs. not exposed) using a chi-squared test. We also tested the bivariate relationship between above-LOD exposure and each of the following variables: number of parental smokers (both or one), child sex, parental nationality, maternal education, paternal education, and income, using chi-squared tests, and child age, parental combined cigarettes per day (CPD), and neighborhood SES, using t-tests. Next, we conducted an adjusted logistic regression analysis to examine whether TSE differed by location of smoking and to identify any correlates that were associated with TSE while controlling for batch. Finally, we ran parallel analyses to those above using the continuous outcome (log hair nicotine) using a multivariable linear model while controlling for batch. We also performed a sensitivity analysis using Tobit regression. 

## 3. Results

### 3.1. Table 1 Describes the Sample Characteristics of Participating Families for Both Studies

In Study 1 (smoking families), the majority of participating families were recruited via Facebook (57.9%), 18.2% were recruited by the snowball method, 15.7% via daycare centers, and 8.2% via advertising. Of the 392 families who expressed initial interest, we were able to contact 357. Of these, 98 families declined to participate after hearing more details; 20 families set a date for a visit but subsequently cancelled the meeting; and 80 families did not meet the inclusion criteria. Thus, 159 families were successfully recruited. 

**Table 1 ijerph-20-03523-t001:** Sample characteristics of participating families.

		Study 1: Smoking Households (*n* = 159)		Study 2: Non-Smoking Households (*n* = 20)	
		*n*	%	*n*	%
Geographic Region	Tel Aviv	63	39.6	6	30
	Central Region	50	31.4	12	60
	Jerusalem	7	4.4	1	5
	Haifa	7	4.4	0	0
	South	30	18.9	1	5
	Judaea and Samaria	2	1.3	0	0
Child sex	Female	82	51.6	8	40.0
	Male	77	48.4	12	60.0
Child age (months)	Mean (SD)	154	37.0 (23.3)	20	29.4 (19.9)
Smoking parent	Neither	0	0.0	20	100.0
	Mother Only	27	17.0	0	0.0
	Father Only	61	38.4	0	0.0
	Both Father and Mother	71	44.7	0	0.0
Parents’ nationality	Both Israeli born	102	64.2	17	85.0
	Other	57	35.8	3	15.0
Mother’s education	Junior/High school	26	16.7	1	5.6
	Some post-high school	24	15.4	1	5.6
	Attended/completed university	106	68.0	16	88.9
Father’s education	Junior/High school	45	29.8	0	0.0
	Some post-high school	37	24.5	1	5.6
	Attended/completed university	69	45.7	17	94.4
Income ^a^	Lower	36	23.5	2	10.0
	Average	49	32.0	3	15.0
	Above	68	44.4	15	75.0
Neighborhood SES ^b^	Mean (SD)	149	13.2 (3.4)	19	14.4 (2.2)

^a^ Based on self-report of greater, the same as, or less than national monthly average household income. ^b^ Based on Israel Central Bureau of Statistics; measured on a scale of 1–20 (20 highest).

In Study 2, most participants were recruited by word of mouth (60.0%), followed by daycare centers (30.0%), and Facebook (5.0%). We do not have a record of the recruitment method for 1 participant. An analysis of hair nicotine was done for 20 children. 

### 3.2. Characteristics of Smoking and Non-Smoking Families

Among non-smoking families, 3 of the 20 families reported that smoking sometimes took place on porches, but none of them reported smoking occurring in any other setting.

### 3.3. Laboratory Batch Results

For the ten batches analyzed, LOD (ng/mg) ranged from 0.014–0.18 (mean LOD: 0.034, median LOD: 0.038). One batch had an LOD of 0.18 and another of 0.09. All other batches had LODs between 0.01 and 0.05. A histogram of hair mass can be found in Appendix A. Hair from the smoking families was analyzed in 9 of the 10 batches, and hair from the non-smoking families was analyzed in 7 of the 10 batches. In total, 141 children from the smoking families and 17 children from the nonsmoking families had sufficient hair for valid laboratory analysis.

### 3.4. Child TSE in Smoking vs. Nonsmoking Families

Of the 141 children from smoking families with sufficient hair, 68.8% (*n* = 97) were exposed (had measurable hair nicotine). Of the 17 children from nonsmoking families with sufficient hair, 35.3% (*n* = 6) were exposed. The difference between them was statistically significant (*p* = 0.006). The mean raw hair nicotine concentration was 0.48 ng/mg (standard deviation (SD) 0.79 in smoking families) and 0.26 ng/mg (SD 0.44) in non-smoking families. The mean log of hair nicotine was −1.85 (SD 1.72) in smoking families and −2.46 (SD 1.54) in non-smoking families.

In the linear regression of LHN, which included a term for laboratory batch, the group variable was borderline significant (*p* = 0.057), batch was statistically significant (*p* < 0.001), and the model R^2^ was 0.30. 

### 3.5. The Association between Child TSE and Location in Smoking Families

Figure 1 presents the proportion of children exposed in each of the home-smoking locations, and Appendix A presents the bivariable analyses of demographic and other variables. Table 2 and Table 3 present the multivariable models for above and below LOD and LHN, respectively.

The most common location where smoking usually took place was on the balcony (39.0%, *n* = 55), followed by smoking in the garden, yard, or other place outside of the home (29.8%, *n* = 42), and smoking limited to designated areas in the home (29.1%, *n* = 41). Few families smoked unrestrictedly throughout the entire house (2.1%, *n* = 3). 

Among children from smoking families, 75.0% of those whose parents smoked in the house were exposed (DSAs: 73.2%, *n* = 41; anyplace in the home: 100%, *n* = 3), while 61.8% of children whose parents restricted smoking to the porch (*n* = 55) were exposed, and 71.4% of those whose parents smoked outside the home (including gardens and yards) (*n* = 42) were exposed. The proportion of children exposed did not differ significantly by smoking location in either the bivariable (*p* = 0.38) or the multivariable analysis for (1) balconies vs. indoors (DSA/anywhere): aOR: 0.74 [0.25–2.16]; or (2) gardens, yards, or other places outside of the home vs. indoors: aOR: 1.26 [0.38–3.92] (*p* = 0.65). Likewise, in the analysis of LHN, location of smoking was not statistically significant in Model 3, which examined the response variables in isolation while controlling for batch (*p* = 0.46) or Model 4, the multivariable model (LSMEAN yard: −2.42, LSMEAN balcony: −2.56, LSMEAN indoors (specified or whole house): −2.41, *p* = 0.87). 

### 3.6. The Association between Child TSE and Other Variables in Smoking Families

The only variable to reach statistical significance in the bivariable comparisons of the proportion of children exposed (Model 1) was paternal education (*p* = 0.001). Of the families where the fathers lacked an academic education, 81.9% of the children were exposed, while among families where the fathers had an academic education, 54.0% of the children were exposed. Paternal education was likewise the only significant potential confounder in the multivariable analysis (Model 2) of proportion exposed (aOR: 5.96 [2.00–17.81], *p* = 0.001). In the multivariable analysis of LHN (Model 4), paternal education was statistically significant (not academic; LSMEAN: −1.97; academic: −2.96; *p* = 0.002). Additionally, in the multivariable analysis of LHN, a greater number of cigarettes smoked by parents was associated with higher exposure (B = 0.034, SE = 0.016, *p* = 0.036). In the sensitivity analysis using Tobit regression, location was not statistically significant (*p* = 0.84). 

## 4. Discussion

The purpose of this study was to compare tobacco smoke exposure [TSE] among children who live in smoking families versus those who do not, as well as among children who live in smoking households with varying smoking locations. We found that about a third of children of non-smoking parents were measurably exposed to tobacco smoke, and roughly double that proportion were exposed in smoking families. Among smoking families, a majority of children in all locations were exposed to detectable TSE, even when parents limited smoking to a porch, a designated room in the house, a window, or the yard. The location of smoking in and around the home was not significantly associated with child exposure, implying that there is no smoking location which reliably protects children from TSE in many Israeli homes.

The finding that about a third of children in non-smoking families were exposed raises questions about how this exposure occurred. Most of the families (85%) in this small sample reported that smoking was not allowed anywhere in or around the home, including indoors, on porches, in yards, or in gardens. Exposure may have taken place via numerous potential routes, such as tobacco smoke incursion from neighboring apartments [32], individuals entering the home with residual smoke on them [33], thirdhand smoke that was deposited in the non-smoking homes by prior occupants [33,34], and/or exposure outside the home, at other people’s homes, in cars, or in public places.

The greater proportion of children exposed to tobacco smoke in smoking families relative to non-smoking families indicates that parental smoking has a key role in child exposure, regardless of where it occurs, and that parental cessation should be encouraged. The exposure is likely caused by a combination of factors, including second- and third-hand exposure. Parents may smoke in close proximity to their children [see Appendix A; smoking proximity is linked to increased child exposure [12]. Air quality inside the home, as measured by PM_2.5_ (small respirable particles), can last up to 5 h [35]. Misconceptions about how and when exposure occurs can lead parents to inadvertently expose their children [13]. Misconceptions can occur because 85% of tobacco smoke is invisible [36] and the sense of smell is unreliable [37]. An additional mechanism for exposure is third-hand smoke, which is the residual smoke that remains after the cigarette is extinguished. [33] Thirdhand smoke exposure is caused by inhalation and by ingestion through the mouth or skin. Thirdhand smoke can remain in the home environment for days, weeks, or months and seep slowly back into the air over time, causing child exposure long after the cigarettes are extinguished or smokers leave [33]. Matt et al. [15] found that infants living in smokers’ homes were exposed to tobacco smoke through air, dust, surfaces, and skin. They also found that levels of nicotine on the fingers of mothers who smoked inside versus outside the home were similar, suggesting that smokers themselves are a vehicle for bringing toxins to the infant, regardless of restrictions on smoking location. A study in a neonatal intensive care unit showed nicotine and carcinogenic tobacco-specific nitrosamine contamination (NNK) in children of smokers and nicotine on the furniture of their hospital rooms, also suggesting that contamination is transferred by the parents themselves [38]. 

Most families in our study (99/141 = 70.2%) allowed smoking on the porch. We found that 61.8% of children from smoking families where home smoking was restricted to the balcony had measurable TSE. Similarly, Bahceciler et al. in Turkey found that 67.6% of children whose parents restricted smoking to outdoor porches were exposed [21]. Children may be directly exposed if they are with their parents on porches when the parents are smoking; parents could behave this way because they mistakenly believe that this protects children from exposure [13]. In the case of completely indoor or partially enclosed porches, this could lead to very high levels of contamination. Cameron et al. found that an overhead cover in an outdoor space increased average exposure by about 50% [39]. Lopez et al., in a study of secondhand smoke exposure on terraces and outdoor areas of hospitality venues, found that the highest levels of respirable particles (PM_2.5_) in all outdoor areas were found in partially enclosed outdoor areas [40].

Indeed, many porches in Israel are entirely or partially within the perimeter of the apartment itself (see Appendix A, Porch A, and Porch C). Porch doors may often be left open in Israel’s climate, which is characterized by hot summers and mild winters. A qualitative study in Israel showed that some smoking parents described smoking on the balcony and leaving the door to the living room open [14]. Outdoor porches that are directly adjacent to the apartment may be small and completely enclosed, semi-enclosed, or open (see Appendix A). Tobacco smoke may accumulate quickly and enter the home itself or a neighbor’s home.

Smoke drift plays an important role in contamination. Sureda et al. conducted a systematic review of SHS in open and semi-open settings and concluded that high SHS levels were present in smoke-free indoor settings that were adjacent to open or semi-open smoking areas [41]. The closer the smoking is to indoor areas, the higher the contamination of the indoor air as measured by PM_2.5_, a standard air quality measure. Other factors influencing smoke levels included wind speed and direction, as well as partial enclosure of outdoor spaces. Brennan et al. found that smoking in adjacent outdoor areas may compromise the air quality of indoor smoke-free areas [42]. Edwards and Wilson concluded that “where free communication exists between outdoor smoking areas and indoor areas, SHS drift can often greatly reduce indoor air quality throughout the pub or bar” [43].

Living conditions in Israel may have affected the exposure of children from both smoking and nonsmoking families. Israel has a very high population density, and most people (74%) live in multi-unit apartment buildings [44]. Crowding in the home, small dwelling sizes, and community smoking levels [45] may contribute to high exposure levels.

Because tobacco smoke incursion from outside the home affects many Israelis, even parental cessation will not completely protect children. An Israeli cross-sectional study showed that 44% of Israelis have experienced tobacco smoke incursion into their own homes, with those of low socio-economic status being disproportionally affected [46].

This study has some limitations. First, the study of smoking families, though it included objective TSE assessment through biomarkers in 141 children, had limited power to detect small differences in exposure. The study of nonsmoking families was small and did not allow multivariable analyses. Second, it is possible that social desirability bias affected the parents’ reporting regarding smoking in the home, resulting in misclassification of the location of smoking even though the interviewers were present in the home at the time of the interview. Third, we did not have information on the smoking habits of visitors (frequent or not) to the home or on whether the children regularly spent time outside of the home with people who smoked or in places where smoking was common, including in cars. We did not have information on the possible confounders of tobacco smoke incursion from nearby apartments, crowding in the home, or small dwelling size. Neither sample was based on a random sample of the population. Finally, it is likely that the effect of smoking in the vicinity of the home on child TSE differs in different areas due to different architectural customs and local climates.

### Public Health Implications

In order to protect children from tobacco smoke exposure, consistent with current scientific evidence on smoke drift outdoors and indoors, smoking should not take place anyplace within at least 9 m [47] of the home, a child, or in cars where children may be present. Porches that are indoors, partially indoors, enclosed, semi-enclosed, or directly adjacent to indoor living areas should be considered part of the home environment. People should refrain from smoking in these areas.

Parents will need substantial support to be able to make changes to better protect their children. We recommend intensive campaigns to normalize the distancing of smoking from homes and children. This information should be included in tobacco package inserts, which have been regulated but not yet implemented [48]. Further, recommendations for the temporary use of nicotine replacement therapy to maintain smoke-free homes, which have been made by the U.K. National Institute for Clinical Excellence (NICE) [49] and Health Scotland [50], should be considered for adoption and subsidized by the National Basket of Health Services.

The governmentally approved “Plan for the Decrease of Smoking and Smoking-attributable Harms in Israel” [51], which was approved by the Israeli Cabinet in 2011, must be updated to produce and swiftly implement a comprehensive tobacco control strategy. Such a strategy must include special consideration for the importance of protecting all children from tobacco smoke exposure, whether it be from their own parents, relatives, neighbors, or others.

## 5. Conclusions

Most children in participating smoking families were exposed to tobacco smoke regardless of whether the parents restricted smoking to a balcony, to designated indoor places such as “in the window,” or to the yard. To protect children from tobacco smoke in the Israeli setting, indoor home porches and outdoor porches that are small and adjacent to indoor living areas, or enclosed or partially enclosed, should be considered part of the home environment and subject to full smoking bans. Full protection of children from the damaging effects of tobacco smoke exposure is likely dependent on reducing smoking prevalence among parents in particular and society as a whole and on complete denormalization of smoking around others, especially vulnerable populations such as children.

## Figures and Tables

**Figure 1 ijerph-20-03523-f001:**
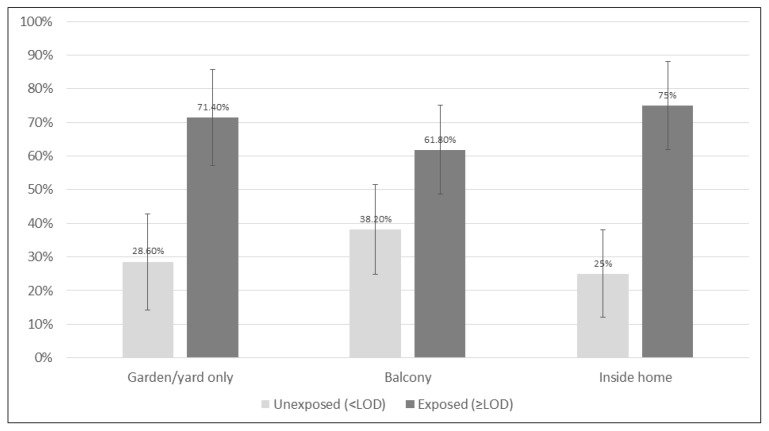
The proportion of children exposed to tobacco smoke as determined by hair nicotine (based on the limit of detection (LOD) of nicotine) by the usual location of smoking. Error bars denote a 95% confidence interval.

**Table 2 ijerph-20-03523-t002:** Logistic regression analyses examining child exposure as assessed by hair nicotine (exposed vs. unexposed, based on Level of Detection (LOD)), by usual location of smoking.

		Odds Ratio	95% CI	*p*-Value
**Model 1 (Bivariable)** *n* = 141	**Location**			0.34
	GYO ^a^ vs. indoors	0.83	0.32–2.17	
	Balcony vs. indoors	0.54	0.23–1.29	
	GYO ^a^ vs. balcony	1.54	0.65–3.66	
	**Batch**	Not estimable		0.66
**Model 2 (Multivariable)** *n* = 115	**Location**			0.65
	GYO ^a^ vs. indoors	1.26	0.38–3.92	
	Balcony vs. indoors	0.74	0.25–2.16	
	GYO ^a^ vs. balcony	1.65	0.56–4.88	
	**Batch**	Not included		
	**Child Age (months)**	0.98	0.96–1.00	0.06
	**Child Sex**			0.50
	Girl vs. Boy	0.73	0.28–1.86	
	**Parental Combined CPD ^b^**	1.03	0.97–1.08	0.33
	**Parental Smokers**			0.65
	Both parents vs. one parent	0.81	0.31–2.07	
	**Parental Nationality**			0.29
	Both Israeli born vs. not both Israeli born	0.58	0.22–1.58	
	**Maternal Education** **^c^**			0.86
	Not academic vs. academic	1.11	0.37–3.30	
	**Paternal Education ^c^**			0.001
	Not academic vs. academic	5.96	2.00–17.81	
	**Monthly household income**			0.20
	Average or below average vs. above average	0.51	0.18–1.42	
	**Neighborhood SES**	1.05	0.90–1.21	0.53

^a^ GYO: garden, yard, or other places outside of the home. ^b^ CPD: cigarettes smoked per day. ^c^ Academic: With at least some university education, versus without any university education.

**Table 3 ijerph-20-03523-t003:** Linear regression analyses examining child exposure as assessed by hair nicotine (log hair nicotine) and by usual location of smoking.

	*n*, Model R^2^		Least Squared Mean/Beta [SE]	*p*-Value
**Model 3 (Bivariable, controlling for batch)**	141, 0.32			
		**Location**		0.46
		GYO ^a^	−2.27	
		Balcony	−2.49	
		Indoors	−2.11	
		Batch	Range:−0.63,−3.86	<0.001
**Model 4 (Multivariable)**	115, 0.43	**Location**		0.87
		GYO ^a^	−2.42	
		LSM Balcony	−2.56	
		LSM Indoors	−2.41	
		**Batch**	Range: −5.06–−0.78	<0.001
		**Child Age (months)**	−0.008 [0.006]	0.21
		**Child Sex**		0.22
		Girl	−2.63	
		Boy	−2.29	
		**Parental Combined CPD ^b^**	0.03 [0.016]	0.04
		**Parental Smokers**		0.30
		Both parents smoke	−2.31	
		One parent smokes	−2.62	
		**Parental Nationality**		0.47
		Both Israeli born	−2.36	
		Not both Israeli born	−2.57	
		**Maternal Education ^c^**		0.42
		Not Academic	−2.59	
		Academic	−2.34	
		**Paternal Education ^c^**		0.002
		Not academic	−1.97	
		Academic	−2.96	
		**Monthly household income**		0.67
		Average or below average	−2.53	
		Above average	−2.40	
		**Neighborhood SES**	0.004 [0.045]	0.92

^a^ GYO: garden, Yard, or other places outside of the home. ^b^ CPD: cigarettes smoked per day. ^c^ Academic: With at least some university education, versus without any university education.

## Data Availability

The data are available from the authors upon reasonable request.

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
