# Peer review of "Tobacco Smoke Exposure According to Location of Home Smoking in Israel: Findings from the Project Zero Exposure Study"

_ijerph, 2023, doi:10.3390/ijerph20043523_

Round 1
Reviewer 1 Report
1. Figure 1 does not have any standard deviation or mean values. Not it is clear what it is about.
2. The y-axis is not lucid, please don't use abbreviations.
3. Lots of grammatical and sentence structure errors. Please ameliorate the manuscript accordingly.
4. Please correct the references section and make it according to the journal's format.
5. Please revisit the linear and multiple regression models and other statistical tools used in the manuscript. They are not properly discussed.
Author Response
- Figure 1 does not have any standard deviation or mean values. Not it is clear what it is about.
- The y-axis is not lucid, please don't use abbreviations.
RESPONSE TO 1&2: Figure 1 presents proportions of exposed children of parents according to location of parental smoking. We have added error bars (95% CIs) and revised the y-axis legend and the caption to improve clarity. - Lots of grammatical and sentence structure errors. Please ameliorate the manuscript accordingly.
RESPONSE: We have carefully reviewed the manuscript and made changes throughout.
4.Please correct the references section and make it according to the journal's format.
RESPONSE: The references have been revised.
- Please revisit the linear and multiple regression models and other statistical tools used in the manuscript. They are not properly discussed.
RESPONSE: We have added text to improve clarity.
Reviewer 2 Report
RE: "Tobacco Smoke Exposure According to Location of Home 2 Smoking: Findings From the Project Zero Exposure Study"
1. The Authors mentioned "Tobacco smoke exposure" among children. Please justify why tobacco smoke exposure was used rather than secondhand tobacco smoke or environmental tobacco smoke. It would be important to clarify (e.g., in the introduction section).
2. The study title should clearly state where this study was carried out, as this study was limited to 1 country (Israel).
3. Please clearly define the study aim (the current sentences at the end of the Introduction require some revisions).
4. Please provide more data on the sampling methods/ methodology of those two projects mentioned in line 107. Please justify why data from two projects were mixed (1-2 sentences will be sufficient).
5. Methods are well-described and this is a strong part of this manuscript.
6. There are only 20 non-smoking households vs. 159 smoking. Please provide an appropriate methodological note (e..g, in Discussion) that will explain the potential concerns related to the sample size.
7. Please provide 2-3 sentences on the practical implications of this study.
Author Response
RE: "Tobacco Smoke Exposure According to Location of Home 2 Smoking: Findings From the Project Zero Exposure Study"
- The Authors mentioned "Tobacco smoke exposure" among children. Please justify why tobacco smoke exposure was used rather than secondhand tobacco smoke or environmental tobacco smoke. It would be important to clarify (e.g., in the introduction section).
RESPONSE: All three terms: tobacco smoke exposure, secondhand smoke exposure, and environmental tobacco smoke are used in the literature. We chose to use tobacco smoke exposure because it includes both second and third hand smoke, both of which contribute to illness and disease. We have added a sentence to clarify that tobacco smoke exposure includes both second and thirdhand smoke. We add that we have used the term tobacco smoke exposure in previous reports of this trial and in other manuscripts. - The study title should clearly state where this study was carried out, as this study was limited to 1 country (Israel).
RESPONSE: The title has been changed accordingly.
- Please clearly define the study aim (the current sentences at the end of the Introduction require some revisions).
RESPONSE: We have revised the aim for clarity (p.3). - Please provide more data on the sampling methods/ methodology of those two projects mentioned in line 107. Please justify why data from two projects were mixed (1-2 sentences will be sufficient).
RESPONSE: These studies were carried out simultaneously, using identical methods, in samples which included smoking families and nonsmoking families. The two studies were not “mixed”, rather exposure levels in the two studies were compared. We have added the sentence: “ Baseline data collection for both studies was identical." (p.3) More details on the methodology of both studies were added to the Methods. - Methods are well-described and this is a strong part of this manuscript.
RESPONSE: Thank you.
- There are only 20 non-smoking households vs. 159 smoking. Please provide an appropriate methodological note (e..g, in Discussion) that will explain the potential concerns related to the sample size.
RESPONSE: In the Limitations Section, we state:
First, the study of smoking families, though it included objective TSE assessment through biomarkers of 141 children, had limited power to detect small differences in exposure. The study of nonsmoking families was small, and did not allow multivariable analyses.
- Please provide 2-3 sentences on the practical implications of this study.
REPSONSE: We added a section on Public Health Implications at the end of the Discussion Section (p.15).
Public Health Implications
In order to protect children from tobacco smoke exposure, consistent with current scientific evidence on smoke drift outdoors and indoors, smoking should not take place anywhere within at least 9 meters[46] of the home, child, or in cars where children may be present. Porches which are indoors, partially indoors, enclosed, semi-enclosed, or directly adjacent to indoor living areas should be considered part of the home environment, with complete smoking bans.
Parents will need substantial support to be able to make changes to better protect their children. We recommend intensive campaigns to normalize distancing of smoking from homes and children. This information should be included in the tobacco package inserts which have been legislated, but not yet implemented. [47] Further, recommendations for temporary use of nicotine replacement therapy to maintain smoke-free homes, which have been made by the U.K. National Institute for Clinical Excellence (NICE)[48] and Health Scotland,[49] should be considered for adoption and subsidized by the National Basket of Health Services.
The governmentally-approved Plan for the Decrease of Smoking and Smoking-attributable Harms in Israel,[50] which was approved by the Israeli Cabinet in 2011, must be updated to produce and swiftly implement a comprehensive tobacco control strategy. Such a strategy must include special consideration for the importance of protecting all children from tobacco smoke exposure, whether it be from their own parents, relatives, neighbors, or others.
Reviewer 3 Report
Review of “Tobacco Smoke Exposure According to Location of Home Smoking: Findings from the Project Zero Exposure Study”
This manuscript presents an analysis of data from a previously published, randomized, interventional study designed to reduce tobacco smoke exposure in the children of smokers, through a behavioral intervention with their parents. The behavioral intervention study included 159 families. Eligibility criteria were that at least one parent smoked 10 or more cigarettes per day and that the child had enough hair to provide a sample sufficient for analysis. The data collected included two hair samples (baseline and 6-8 months later) from one child in each family, as well as data how many cigarettes each parent smoked per day, how often people smoked in the home, where in the home smoking was allowed and how often people smoked in the vicinity of the children.
This manuscript reports demographic and smoking behavior data from the 159 families that participated in the prior study plus an additional 20 nonsmokers recruited for a new study. It includes hair nicotine data from 141 of the families in the prior study plus an additional 17 families from the study of nonsmoking families. The nicotine content of the hair samples collected at the baseline study visits was measured. The relationship between the child’s hair nicotine concentrations at the baseline visit and smoking status of their parents as well as the locations within the home where people smoked were analyzed by logistic and linear regression. The studies were performed in Israel.
The authors found that 68.8% of the children whose parents smoked had hair nicotine concentrations above the limit of detection, while only 35.3% of the children whose parents did not smoke did (p= 0.006). Mean hair nicotine was 0.48 ng/mg hair in smoking families and 0.26 ng/mg in nonsmoking families. They also detected effects on child’s hair nicotine content for the father’s educational level, with more educated fathers having children with lower hair nicotine content. However, they detected no effect on child’s hair nicotine content for location within the household where smoking was allowed. In the discussion section, the authors focus on the fact that limiting smoking to specific parts of the home, like the balcony, did not result in any detectable reduction in exposure. They end the manuscript by recommending that public health campaigns educate parents that it is unsafe to smoke in any part of the house, including balconies and porches.
The largest weaknesses in this paper are the significant differences in the limit of detection for hair nicotine between the 10 sets of samples sent to the analytical laboratory and the lack of integration of the concept of thirdhand smoke on into their analyses of their data.
Higher density may explain why the authors did not find a difference in smoke exposure between the families with different home smoking location policies, but the small number of families with a completely unrestricted home smoking policy and the loss of sensitivity caused by the batch to batch variation in the hair nicotine assay LOD are just as likely to explain this.
The authors sent 10 sets of hair samples to an analytical laboratory at Johns Hopkins University to have the nicotine content measured. They stated “For the 10 batches analyzed, LOD ranged from 0.014-0.18.” I do not understand why the limit of detection for nicotine in hair should vary that widely from run to run at a competent laboratory. The limit of detection for chemical analyses of biological specimens is determined by three factors: the limits of the instrument to detect and resolve a given analyte, the volume of the solvent that the specimen is extracted into, and the mass of the biological specimen. The limits of the instrument are near constant and the volumes of solvent used for digestion of the hair, extraction of the digest and injection into the instrument are normally held constant unless there is a special request for low sample volume to push the limit lower. The primary variable affecting the limit of detection for these analyses should have been the mass of hair. Hair mass varies with the length and diameter of each shaft and the number of hair shafts in the sample. Thus, the limit of detection should vary from sample to sample, not from batch to batch.
In the method paper from the Johns Hopkins researchers that they cited, the LOD was 0.02 ng/mg for a 30 mg hair sample(1). The near 13-fold batch to batch variation reported in this manuscript suggests that the laboratory may have taken the limit of detection for the single hair sample with the lowest mass and applied it to the entire batch. I do not understand why they would do that. It results in a loss of information and has clearly affected the analyses in this manuscript. The laboratories I have worked with in the past provide me with the limit of quantitation for the analyte and instrument that is specific to the solvent volume that the sample is digested in and extracted into and the amount of the solvent that is injected. I then calculate the limit of quantitation for each sample, based on the mass of the sample and report a range of LOQs for the samples. I suggest that the authors communicate with their analytical laboratory and see if they can recover more information from the analyses in this way. The revisions should include an explanation of why the LOD varied so much. The authors should also specify how many mgs of hair were deemed sufficient for the study and include a histogram of all hair sample masses in the supplementary section.
The authors define thirdhand cigarette smoke and cite papers that were foundational in the creation of the term. However, an understanding of how thirdhand smoke can affect nicotine exposure is not integrated into their interpretation and conclusions. For example, in the discussion of why a third of children in nonsmoking homes were positive for nicotine, the authors posit exposures to secondhand smoke that are outside the home or by family members other than the parents. They should add that the source of exposure may have been thirdhand smoke deposited in the nonsmoking homes by prior occupants of the home(2) or smoke that seeps from neighbors’ homes into the nonsmoking home(3).
The last lines of the abstract read “In conclusion, the majority of children in smoking families were measurably exposed to TSE, even if the location of smoking was restricted to designated areas in the home, balconies or yards. This is likely attributable to the indoor our partially enclosed and/or covered nature of many balconies, and the proximity of outdoor porches and yards in Israel to indoor living spaces.” The authors’ focus on the architecture of balconies and porches and the high density of housing in Israel ignores findings in the 2004 paper by Matt et al. (4) that even in an area with lower density, like San Diego, California, and even in families that only smoked outdoors and closed their windows every or almost every time they went out to smoke, the children were exposed to 5-7 times as much nicotine as the children of nonsmokers. A critical finding in that paper was that the hands of all the smoking parents in that study had equal amounts of nicotine on them. A recent study of nicotine and NNK contamination in children in the neonatal intensive care unit of a hospital showed that the children of smokers have nicotine and NNK metabolites in their urine and nicotine on the furniture of their hospital rooms(5). That didn’t come from smoke drift from nearby rooms balconies and it wasn’t because the parents were smoking less than 9 meters from their sick children: the hospital smoking ban was campus-wide and strictly enforced. The nicotine and NNK came from their parents’ bodies, especially their hands. These ideas are summarized and explained in a recent review of research on thirdhand smoke (6).
Higher density may explain why the authors did not find a difference in smoke exposure between the families with different home smoking location policies, but the small number of families with either a completely unrestricted home smoking policy and loss of data caused by the batch to batch variation in the hair nicotine assay LOD are just as likely to explain this.
While it is understandable that the authors want to counter an idea (that smoking on the balcony, a few steps away from the main living space in a flat, is sufficient to protect children from smoke exposure) that may be prevalent in Israel now, their paper will be stronger if they integrate the latest research on thirdhand smoke into their analysis. The recommendation that parents quit smoking shouldn’t be relegated to the last line of the paper. It is the best way to truly protect children from exposure to tobacco smoke toxins.
Minor problems:
It would help if the authors introduced the concept of smoke particles or PM2.5 in the air and how they correlate with nicotine exposure. The work by Sureda et al. relied primarily on PM2.5 measurements and the phrase “the closer to the smoking to the indoor areas, the higher contamination of the indoor air” would read better as “…the higher the PM2.5 concentrations indoors.” Likewise, “Tobacco smoke within a home can remain there for as long as 5 hours” makes no sense because we know that thirdhand smoke can linger for years. It makes better sense as “Particulate matter from tobacco smoke can linger in indoor air for as long as 5 hours.”
The sentence on lines 307-310 does not make sense as written.
Working Bibliography
1. Kim SR, Wipfli H, Avila-Tang E, Samet JM, Breysse PN. Method validation for measurement of hair nicotine level in nonsmokers. Biomed Chromatogr. 2009;23(3):273-9.
2. Matt GE, Quintana PJ, Zakarian JM, Fortmann AL, Chatfield DA, Hoh E, et al. When smokers move out and non-smokers move in: residential thirdhand smoke pollution and exposure. Tob Control. 2011;20(1):e1.
3. Licht AS, King BA, Travers MJ, Rivard C, Hyland AJ. Attitudes, experiences, and acceptance of smoke-free policies among US multiunit housing residents. Am J Public Health. 2012;102(10):1868-71.
4. Matt GE, Quintana PJ, Hovell MF, Bernert JT, Song S, Novianti N, et al. Households contaminated by environmental tobacco smoke: sources of infant exposures. Tob Control. 2004;13(1):29-37.
5. Northrup TF, Khan AM, Jacob P, 3rd, Benowitz NL, Hoh E, Hovell MF, et al. Thirdhand smoke contamination in hospital settings: assessing exposure risk for vulnerable paediatric patients. Tob Control. 2015.
6. Jacob P, 3rd, Benowitz NL, Destaillats H, Gundel L, Hang B, Martins-Green M, et al. Thirdhand Smoke: New Evidence, Challenges, and Future Directions. Chem Res Toxicol. 2017;30(1):270-94.
Author Response
This manuscript presents an analysis of data from a previously published, randomized, interventional study designed to reduce tobacco smoke exposure in the children of smokers, through a behavioral intervention with their parents. The behavioral intervention study included 159 families. Eligibility criteria were that at least one parent smoked 10 or more cigarettes per day and that the child had enough hair to provide a sample sufficient for analysis. The data collected included two hair samples (baseline and 6-8 months later) from one child in each family, as well as data how many cigarettes each parent smoked per day, how often people smoked in the home, where in the home smoking was allowed and how often people smoked in the vicinity of the children.
This manuscript reports demographic and smoking behavior data from the 159 families that participated in the prior study plus an additional 20 nonsmokers recruited for a new study. It includes hair nicotine data from 141 of the families in the prior study plus an additional 17 families from the study of nonsmoking families. The nicotine content of the hair samples collected at the baseline study visits was measured. The relationship between the child's hair nicotine concentrations at the baseline visit and smoking status of their parents as well as the locations within the home where people smoked were analyzed by logistic and linear regression. The studies were performed in Israel.
The authors found that 68.8% of the children whose parents smoked had hair nicotine concentrations above the limit of detection, while only 35.3% of the children whose parents did not smoke did (p= 0.006). Mean hair nicotine was 0.48 ng/mg hair in smoking families and 0.26 ng/mg in nonsmoking families. They also detected effects on child's hair nicotine content for the father's educational level, with more educated fathers having children with lower hair nicotine content. However, they detected no effect on child's hair nicotine content for location within the household where smoking was allowed. In the discussion section, the authors focus on the fact that limiting smoking to specific parts of the home, like the balcony, did not result in any detectable reduction in exposure. They end the manuscript by recommending that public health campaigns educate parents that it is unsafe to smoke in any part of the house, including balconies and porches.
The largest weaknesses in this paper are the significant differences in the limit of detection for hair nicotine between the 10 sets of samples sent to the analytical laboratory and the lack of integration of the concept of thirdhand smoke on into their analyses of their data.
REPSONSE:
1 - Thirdhand smoke concepts have now been added to various sections of the paper. We agree that this is indeed an important route of exposure among young children.
2 - Regarding the differences in the LOD, and to what extent that may have impacted our results: this was a concern for us as well. We did the following:
- We contacted our laboratory, as you suggested. They informed us that i – the LOD used was the mean LOD for that batch, and ii LOD was high in the first and second batch, and was particularly high in the first batch (0.18), coinciding with the greatest number of samples with low hair mass. In subsequent batches we were careful to obtain larger hair samples. Please see detailed responses below. We have also added information to the manuscript (Section 2.4.1, Primary Outcome).
- We conducted sensitivity analyses on the data. These are explained below.
Higher density may explain why the authors did not find a difference in smoke exposure between the families with different home smoking location policies, but the small number of families with a completely unrestricted home smoking policy and the loss of sensitivity caused by the batch to batch variation in the hair nicotine assay LOO are just as likely to explain this.
RESPONSE: The Reviewer proposes two possible alternate explanations for our findings. Regarding the first possible explanation, that there were only 3 families with completely unrestricted home smoking policy, of the 141 with sufficient hair to assess exposure: Parenthetically, we see this as a positive outcome, indicating a certain denormalization of smoking anywhere in the home. Regarding the substance: from the start, for the purposes of analyses, we did not think it reasonable to conduct analyses on cells with just 3 observations. Therefore, we combined the categories of “smoking anywhere” with “smoking inside the home in designated areas” for the purpose of multivariable analyses. There were therefore 44 children included in the location of Smoking in the home. We think this is reasonable, especially considering the sample sizes of other studies on these topics. We did change Figure 1, Percent exposed by smoking location, to include just 3 categories instead of 4; now it is consistent with the statistical analyses. We note that in the manuscript, we did report on percent exposed anywhere in the home, versus at designated areas within the home, because we wanted a number (percent exposed) with which to counter the common misconceptions that “smoking in the window” or “in designated home areas” is protective of children.
The authors sent 10 sets of hair samples to an analytical laboratory at Johns Hopkins University to have the nicotine content measured. They stated "For the 10 batches analyzed, LOO ranged from 0.014-0.18." I do not understand why the limit of detection for nicotine in hair should vary that widely from run to run at a competent laboratory. The limit of detection for chemical analyses of biological specimens is determined by three factors: the limits of the instrument to detect and resolve a given analyte, the volume of the solvent that the specimen is extracted into, and the mass of the biological specimen. The limits of the instrument are near constant and the volumes of solvent used for digestion of the hair, extraction of the digest and injection into the instrument are normally held constant unless there is a special request for low sample volume to push the limit lower. The primary variable affecting the limit of detection for these analyses should have been the mass of hair. Hair mass varies with the length and diameter of each shaft and the number of hair shafts in the sample. Thus, the limit of detection should vary from sample to sample, not from batch to batch.
RESPONSE: From the laboratory: The reviewer is correct that the mass of hair is the primary variable driving LOD, and LOD varies from sample to sample. These analyses were performed in hair from small children, which limited the mass available for analysis. The lab analyzed samples as small as 5 mg in an attempt to minimize censoring, however on the advice of the lab, only samples weighing at least 10mg were included in the analysis, due to uncertainty of smaller samples. The lab reported the mass and the LOD for investigators to consider when performing the statistical analysis.
We would also like to point out that LOD was highest in the first batch, coinciding with the most number of samples with low hair mass – in subsequent batches we were careful to obtain larger hair samples. We have added a histogram of hair mass as a Supplementary file.
In the method paper from the Johns Hopkins researchers that they cited, the LOO was
0.02 ng/mg for a 30 mg hair sample(1). The near 13-fold batch to batch variation reported in this manuscript suggests that the laboratory may have taken the limit of detection for the single hair sample with the lowest mass and applied it to the entire batch. I do not understand why they would do that. It results in a loss of information and has clearly affected the analyses in this manuscript.
RESPONSE: From the laboratory: The laboratory uses the average LOD for each batch. As the reviewer points out, this approach is not ideal, but it’s the approach that was agreed upon for statistical analyses. As a clarifying point, we have gone back to the individual batch reports and verified that the highest LOD of 0.18 applied to only one batch (the first one). One more batch had an LOD of 0.09 and the rest of the batches had LODs between 0.014 and 0.05, with a mean of 0.034 and a median of 0.038. Furthermore, independent of the batch LOD, each individual sample was deemed above or below the LOD based on the mass of nicotine in the sample, which we believe is a good way to assess exposure in the present study.
We have added the mean and median LOD values to the manuscript.
The laboratories I have worked with in the past provide me with the limit of quantitation for the analyte and instrument that is specific to the solvent volume that the sample is digested in and extracted into and the amount of the solvent that is injected. I then calculate the limit of quantitation for each sample, based on the mass of the sample and report a range of LOQs for the samples. I suggest that the authors communicate with their analytical laboratory and see if they can recover more information from the analyses in this way. The revisions should include an explanation of why the LOO varied so much. The authors should also specify how many mgs of hair were deemed sufficient for the study and include a histogram of all hair sample masses in the supplementary section.
RESPONSE: The histogram of the distribution of hair weights appears below and in Supplementary Figure S1.
RESPONSE: We add here that, in addition to the aforementioned clarifications, we performed several sensitivity analyses to address the issue of variations in LOD between batches. First, we excluded all observations from Batch 1 , and from Batch 1 and Batch 2, from the analyses. The p-values for location remained high, with values close to those obtained with Batch 1, and Batch 1 and 2, included. Second, we performed a Tobit regression on LHN. Again, the p-value for Location was very similar to that obtained in the linear regression. We included the Tobit regression in the manuscript.
The similarity of the results of these sensitivity analyses, which are nearly identical to results from the original multivariable and bivariable analyses, on percent exposed and on LHN, gave us confidence that our results are relatively robust to changes in statistical approaches, and that they do reflect a lack of substantial effect due to smoking location. The very high p-values which we found for the Location variable (p=.65 for Location variable in multivariable logistic regression, p=.87 for the multivariable linear regression, p=.8405, Tobit regression) suggest that this is not a simple issue of low power due to limited sample size: were that the case, the p-values would have been much closer to 0.05.
The authors define thirdhand cigarette smoke and cite papers that were foundational in the creation of the term. However, an understanding of how thirdhand smoke can affect nicotine exposure is not integrated into their interpretation and conclusions. For example, in the discussion of why a third of children in nonsmoking homes were positive for nicotine, the authors posit exposures to secondhand smoke that are outside the home or by family members other than the parents. They should add that the source of exposure may have been thirdhand smoke deposited in the nonsmoking homes by prior occupants of the home(2) or smoke that seeps from neighbors' homes into the nonsmoking home(3).
RESPONSE: Thank you, we have added these points and the references.
The last lines of the abstract read "In conclusion, the majority of children in smoking families were measurably exposed to TSE, even if the location of smoking was restricted to designated areas in the home, balconies or yards. This is likely attributable to the indoor our partially enclosed and/or covered nature of many balconies, and the proximity of outdoor porches and yards in Israel to indoor living spaces." The authors' focus on the architecture of balconies and porches and the high density of housing in Israel ignores findings in the 2004 paper by Matt et al. (4) that even in an area with lower density, like San Diego, California, and even in families that only smoked outdoors and closed their
windows every or almost every time they went out to smoke, the children were exposed to 5-7 times as much nicotine as the children of nonsmokers. A critical finding in that paper was that the hands of all the smoking parents in that study had equal amounts of nicotine on them.
RESPONSE: We have substantially revised our discussion and conclusion sections to include this information.
A recent study of nicotine and NNK contamination in children in the neonatal intensive care unit of a hospital showed that the children of smokers have nicotine and NNK metabolites in their urine and nicotine on the furniture of their hospital rooms(5).
That didn't come from smoke drift from nearby rooms balconies and it wasn't because the parents were smoking less than 9 meters from their sick children: the hospital smoking ban was campus-wide and strictly enforced. The nicotine and NNK came from their parents' bodies, especially their hands. These ideas are summarized and explained in a recent review of research on thirdhand smoke (6).
RESPONSE: Agreed. Thank you for introducing us to these studies. We have added this information to the paper. We particularly enjoyed the paper by Jacob: we had been unaware of that paper, and it is a wonderful summary of the literature.
Higher density may explain why the authors did not find a difference in smoke exposure between the families with different home smoking location policies, but the small number of families with either a completely unrestricted home smoking policy and loss of data caused by the batch to batch variation in the hair nicotine assay LOO are just as likely to explain this.
RESPONSE: We answered these issues above in relation to a previous comment.
We think the loss of data due to batch variation is probably not true, because we adjusted for batch in the analyses.
While it is understandable that the authors want to counter an idea (that smoking on the balcony, a few steps away from the main living space in a flat, is sufficient to protect children from smoke exposure) that may be prevalent in Israel now, their paper will be stronger if they integrate the latest research on thirdhand smoke into their analysis. The recommendation that parents quit smoking shouldn't be relegated to the last line of the paper. It is the best way to truly protect children from exposure to tobacco smoke toxins.
RESPONSE: Thank you , we agree and have incorporated these ideas into the paper.
Minor problems:
It would help if the authors introduced the concept of smoke particles or PM2.5 in the air and how they correlate with nicotine exposure. The work by Sureda et al. relied primarily on PM2.5 measurements and the phrase "the closer to the smoking to the indoor areas, the higher contamination of the indoor air" would read better as "...the higher the PM2.5 concentrations indoors."
RESPONSE: We have made the change.
Likewise, "Tobacco smoke within a home can remain there for as
long as 5 hours" makes no sense because we know that thirdhand smoke can linger for
years. It makes better sense as "Particulate matter from tobacco smoke can linger in indoor air for as long as 5 hours."
RESPONSE: We have modified this.
The sentence on lines 307-310 does not make sense as written.
RESPONSE: This has been revised. We hope that it is clearer now.
THANKYOU VERY MUCH FOR RAISING IMPORTANT ISSUES AND PROVIDING INFORMATION ON IMPORTANT RELEVANT LITERATURE.
Working Bibliography
- Kim SR, Wipfli H, Avila-Tang E, Samet JM, Breysse PN. Method validation for measurement of hair nicotine level in nonsmokers. Biomed Chromatogr. 2009;23(3):273-9.
- Matt GE, Quintana PJ, Zakarian JM, Fortmann AL, Chatfield DA, Hoh E, et al. When smokers move out and non-smokers move in: residential thirdhand smoke pollution and exposure. Tab Control. 2011;20(1):e1.
- Licht AS, King BA, Travers MJ, Rivard C, Hyland AJ. Attitudes, experiences, and acceptance of smoke-free policies among US multiunit housing residents. Am J Public Health. 2012;102(10):1868-71.
- Matt GE, Quintana PJ, Hovell MF, Bernert JT, Song S, Novianti N, et al. Households contaminated by environmental tobacco smoke: sources of infant exposures. Tab Control. 2004;13(1):29-37.
- Northrup TF, Khan AM, Jacob P, 3rd, Benowitz NL, Hoh E, Hovell MF, et al. Thirdhand smoke contamination in hospital settings: assessing exposure risk for vulnerable paediatric patients. Tab Control. 2015.
- Jacob P, 3rd, Benowitz NL, Destaillats H, Gundel L, Hang B, Martins-Green M, et al. Thirdhand Smoke: New Evidence, Challenges, and Future Directions. Chem Res Toxicol. 2017;30(1):270-94.
Round 2
Reviewer 2 Report
This study may be accepted for publication.
Author Response
Thank you.
Reviewer 3 Report
The manuscript is substantially improved and I'm glad that you found the additional literature helpful.
There are two minor changes that still need to be made:
On line 95, the authors state “smoking anywhere would be associated with the highest levels of TSE”. Please provide a definition for “anywhere”. Something like “an unrestricted smoking policy, where smoking is allowed in any room or part of the home, would be associated with the highest levels of TSE”.
On line 220, remember to give the units for the LOD: ng/mg hair.
Author Response
The changes have been made as requested.
Line 96 now reads: We hypothesized that an unrestricted smoking policy, where smoking is allowed in any room or part of the home
Line 216 now reads: For the ten batches analyzed, LOD (ng/mg) ranged from 0.014-0.18